# Comparison of Efficacy of Fermented Garlic and Orlistat (Lipase Inhibitor) in Obesity Management Using an Experimental Rodent Model

**DOI:** 10.3390/foods12213905

**Published:** 2023-10-25

**Authors:** Mavra Javed, Waqas Ahmed, Azmatullah Khan, Imtiaz Rabbani

**Affiliations:** 1Department of Food Science and Human Nutrition, University of Veterinary and Animal Sciences, Lahore 54000, Pakistan; 2Department of Physiology, University of Veterinary and Animal Sciences, Lahore 38040, Pakistan

**Keywords:** fermented garlic, orlistat, high-fat diet, animal model, anti-obesity

## Abstract

Background: Black garlic, also known as fermented garlic, is a useful food that may have therapeutic benefits. The aim of this study was to analyze the impact of fermented garlic and orlistat therapy on obese rats. Methods: A total of 40 male albino rats (245–250 g) were fed either an HFD (*n* = 32) or a normal diet (*n* = 8) for 6 weeks; therefore we randomly assigned the rats into: group I (normal diet), group II (HFD), groups III and IV (HFD with fermented garlic), and group V (orlistat for) 6 weeks. Two different dosages of fermented garlic (481.2 mg/kg and 963.3 mg/kg) were administered. Afterward, blood was collected, body weight was measured, and tissue was collected for further analysis. Results: Both the orlistat and black garlic groups showed a significant reduction in BMI, lipid profiles, and insulin levels compared with the baseline. The orlistat group showed significant elevation (*p* < 0.005) in body weight, organ weight, lipids, and liver parameters, with histopathological findings. The administration of black garlic improved the inflammatory markers with all other parameters. Conclusion: The fermented garlic and orlistat reinstated all of the investigated parameters significantly (*p* < 0.05), especially body weight and lipid profiles, and induced histopathological changes compared to the drug orlistat. Additionally, it showed anti-obesity-related therapeutic impacts compared with the orlistat drug. Black garlic provides a reliable and effective treatment for obesity compared to orlistat.

## 1. Introduction

The predominance of obesity, a global health concern, has been growing in developing nations because of food and lifestyle changes. Obesity is a prominent reason for disease and fatality. It is difficult to address due to its association with increased risk of metabolic syndrome, cardiovascular diseases, heart attack, hypertension, diabetes mellitus, and different types of cancer [1] According to the World Health Organization, about 1.9 billion people are overweight, with over 650 million of them being considered obese [2]. Psychological therapy, environmental improvements, and education are persistently used methods for evading obesity [3]. In recent years, attention has been drawn to the usage of anti-obesity medications to cut body weight by reducing portions of food or inhibiting absorption, or by increasing energy expenditure [4]. The existing anti-obesity drugs have harmful side effects, such as cardiovascular diseases with stroke in heart patients, pulmonic hypertension, and psychological outcomes [5]. Orlistat (also known as Xenical and Alli), a drug certified by the Food and Drug Administration (FDA), can be used throughout life for weight management. Orlistat, a lipase inhibitor, has been demonstrated to be helpful for both weight loss and weight maintenance. It obstructs dietary fat absorption by about 30%. Despite the FDA’s approval of orlistat, it has been associated with gastrointestinal issues such indigestion, bloating, abdominal pain, diarrhea, and flatulence [6]. Recently, reports of serious liver damage have also been recounted (gov, 2017). Garlic (*Allium sativum* L.) has anti-obesity, antibacterial, antihypertensive, blood glucose-lowering, antithrombotic, antimutagenic, and antiplatelet effects, among other pharmacological effects [7,8,9,10,11]. Garlic oil and the organosulfur compounds it contains have been revealed to be useful for preventing rats on high-fat diets (HFDs) from increasing in weight [12]. In addition, bioconversion technology alters a variety of organic compounds by using specific procedures to convert them into certain substances for dedicated purposes. Garlic consumption has been underappreciated because of its overpowering flavor and scent, which might cause stomach discomfort. On the food market, alternatives to raw garlic have been produced, including black garlic, garlic powder, and chemically processed garlic. In Mie Prefecture, black garlic was initially found by Japanese scientist Mr. Kamimura. It has a pleasant fruit-like taste without a pungent scent and is simple to use [13]. Black garlic, also known as aged garlic, is a form of fermented garlic made from raw garlic. This product is utilized in Asian countries, including Taiwan (China), Japan, and Korea, as a food component and a functional food [14]. Black garlic is made by thermally aging raw garlic at high temperatures and high humidity. As a result of these processes, the volatile and pungent chemicals in raw garlic are either transformed into stable and odorless molecules, such as S-allyl cysteine (SAC), or degraded into organosulfur compounds such as dithiins and ajoene [15]. It has been found that black garlic and its constituent reduce oxidative stress, inflammatory markers, and elevated plasma concentrations [16]. It is also reported that black garlic can be used for the improvement of hyperlipidemic profiles [17]. It has been established that black garlic extract has positive effects on lowering cholesterol and preventing weight gain and visceral fat growth in rats consuming a high-fat diet [18]. It has been indicated that consuming fermented black garlic may help reduce diabetes complications caused by a high-fat diet [19].

The objective of this research is to evaluate the anti-obesity effect of fermented garlic in addition to biochemical parameters, organ and adipose tissue weight, and histology in a rat model fed a high-fat diet to account for the side effects of orlistat.

## 2. Materials and Methods

### 2.1. Materials

Desi garlic was purchased from a local market. Orlistat drug was purchased from the pharmaceutical company Pharm Evo Private Limited (Karachi, Pakistan) under the market name Orslim^®^ (Karachi, Pakistan). Other chemicals and reagents for testing were purchased Sigma-Aldrich, Co (St. Louis, MO, USA).

### 2.2. Preparation of Fermented Garlic Powder

Garlic cloves were purchased from a local market. They were cleaned of any dirt and extra skin. Black garlic was produced by following the method mentioned in [20] with some modifications. The garlic was placed in a temperature-controlled chamber at 70 °C and 90% relative humidity for 45 days. This garlic was stored in a −20 °C for further use. Black garlic powder was prepared following the method mentioned in [21]; 100 g of black garlic was chopped and ground into a fine form, and then, it was dried in a hot air oven at 60 °C for 24 h. It was ground to a fine powder, and then, stored at −80 °C until required for experimentation.

### 2.3. Animal Experiment

In this study, 40 male albino rats with bodyweights in the range of 245–250 g were purchased from the Institute of Biochemistry at the University of Veterinary and Animal Sciences, Lahore. The trial rats remained in animal houses for 7 days and were kept in a 12:12 h light/dark cycle at 20 °C ± 2 for 14 weeks with food and water. The study was approved by ethical review committee of the University of Veterinary and Animal Sciences, Lahore (No. DR/418 dated 13 October 2021). The high-fat diet was made by mixing 30% (*w*/*w*) lard into a normal diet to induce obesity in the experimental animals. The normal feed contained 49% carbohydrates, 21% protein, 3% fat, 0.8% calcium, 0.4% phosphorus, 5% fiber, 13% moisture, and 8% ash. The fat content of the high-fat diet was 33%.

### 2.4. Experimental Design

Rats were distributed into 5 groups, with 8 rats in each group. Group I was the control group with a normal diet, while the other groups were fed a high-fat diet for 8 weeks. After 8 weeks of high-fat diet, the animals were divided into 4 groups for supplementation with orlistat and fermented garlic powder. The animals were divided into the following groups and the experiment continued for 6 weeks: group II (high-fat diet), group III (high-fat diet + fermented garlic powder level 1), group IV (high-fat diet + fermented garlic powder level 2), and group V (high-fat diet + orlistat). The calculated dosages were given to the groups along with a high-fat diet. The animals were weighed every day during the trial. Organ weight was also noted after the end of trial. Blood was drawn from each group before and after the treatment to assess the effect of orlistat and fermented black garlic. The dosage of fermented black garlic and orlistat was calculated using the human equivalent dosage (HED).
(1)$$\mathrm{HED}\left(\frac{\mathrm{mg}}{\mathrm{kg}}\right)=\mathrm{Animal}\;\mathrm{dose}\;\left(\frac{\mathrm{mg}}{\mathrm{kg}}\right)\times \frac{{\left(\mathrm{animal}\;\mathrm{weight}\;\mathrm{in}\;\mathrm{kg}\right)}^{0.33}}{\mathrm{Human}\;\mathrm{weight}\;\mathrm{in}\;\mathrm{kg}}$$

The calculated dosages of fermented garlic for groups III and IV were 481.2 mg/kg and 963.3 mg/kg, and the dosage of orlistat was 43.3 mg/kg, according to the human equivalent dose level.

### 2.5. Collection of Blood Samples

Prior to the 6 weeks of treatment, blood was drawn from the heart of the animals under chloroform anesthesia. For the estimation of liver and lipid profile parameters, serum was separated from blood using a centrifuge machine at 1500 rpm for 10 min. Towards the end of the experiment, the animals were fasted overnight and sacrificed via cervical dislocation under chloroform anesthesia. Blood was taken via cardiac puncture into sterile tubes and allowed to stand for 30 min at 20–25 °C. The rats were immolated. Liver and adipose tissues were removed right away and dipped in chilled physiological saline. Tissues were marked, weighed precisely, and placed in formalin solution.

### 2.6. Blood Serum Lipid Profiles and Insulin

The lipid profiles, including the total cholesterol, LDL, VLDL, HDL, triglycerides, and insulin in serum, were assessed using commercial kits purchased from Sigma-Aldrich using a UV spectrophotometer with absorbance at 505/670 nm. All the results are presented in mg/dL.

### 2.7. Blood Serum Liver Function Parameters

The liver function parameters, including aspartate transaminase (AST), alanine transaminase (ALT), alkaline phosphates (ALP), albumin (ALB), bilirubin, and the AG ratio, were determined by following the methods mentioned in the kits (Sigma-Aldrich)

### 2.8. Serum IL-6 and C-Reactive Protein

The plasma levels of serum IL-6 and C-reactive protein were also assessed using ELISA kits.

### 2.9. Histopathology Slides

Rats were sacrificed at the end of the conducted experiment via heart puncture. All blood was washed from the circulation by rinsing it with saline until the fluid turned clear; then, it was flushed with 10% formal solution for fixation. The formaldehyde-fixed tissues were handled for routine paraffin sectioning and stained with Hematoxylin and Eosin (H&E). Briefly, tissues in a graded series were hydrated and dehydrated, dipped in chloroform and xylene, and then, fixed in paraffin wax. Using a rotary microtome, tissues were cut into 5 mm thick sections, after which they were left at room temperature overnight. Following deparaffinization, the sections underwent a declining alcohol series of rehydration (100% alcohol, 90% alcohol, 70% alcohol, and 50% alcohol), and then, were rinsed with distilled water. These sections were quickly passed through an increasing alcohol series after being stained with H&E. A Nikon (40 × 10) microscope was used to examine and record the morphology of the tissues.

### 2.10. Statistical Analysis

Data were statistically analyzed using the Statistical Package for Social Sciences (SPSS) IBM, SPSS Inc, USA version 27.0 for windows. Body weight data were analyzed via repeated measures two-way ANOVA. Differences in organ weights were also assessed via analysis of covariance (ANCOVA) to determine if they are explained by variations in body weight. One-way analysis of variance (ANOVA) was used to compare the standard error of the means. Values were considered significant at *p* < 0.05. Inter-group comparisons were performed using Tukey’s post-hoc test and the Dunnett test. GraphPad prism 8.0.0 software (San Diego, California) as used for graphical configuration.

## 3. Results

### 3.1. Body and Organ Weight

In this study, the high-fat diet group (30% *w*/*w* lard) was divided in to four groups. Black garlic with two levels of the drug orlistat was administered at different dosages according to the human equivalent dosage for six weeks. In Table 1, the two-way analysis between weeks (F= 496.88; *p* < 0.001) and groups (F = 153.79; *p* < 0.001) showed a significant effect. Post-hoc analysis further explained the results, as group II showed a significant increase in weight in the following weeks compared to the control (*p* < 0.001). Fermented garlic treatment groups III and IV also showed a significant decrease in weight compared to the control (*p* < 0.001). Group V showed a significant decrease in weight compared to the control (*p* < 0.001).

The weights of the organs and tissues are shows in Figure 1. Analyzing the weights of livers (F = 2955.17; *p* < 0.001), adipose tissue (F = 1682.9; *p* < 0.001), soleus muscles (F = 236.8; *p* < 0.001), and gastrocnemius muscles (F = 390.3; *p* < 0.001) of different groups via one-way analysis showed significant changes. Further post-hoc analysis showed that group II had a significant increase in the weight of liver (*p* < 0.001), adipose tissue (*p* < 0.001), soleus muscle (*p* < 0.001), and gastrocnemius muscle (*p* < 0.000). All of group V (orlistat) had a more significant decrease in the weight of the liver (*p* < 0.001), adipose tissue (*p* < 0.001), soleus muscle (*p* < 0.001), and gastrocnemius muscle (*p* < 0.001) compared to the control. Groups III and IV showed a significant decrease in organ and tissue weight (liver (*p* < 0.001), adipose tissue (*p* < 0.001), soleus muscle (*p* < 0.001), and gastrocnemius muscle (*p* < 0.001)) compared to the control group.

### 3.2. Serum Lipid Profile

Changes in the lipid profile parameters are shown in Table 2. Serum high-fat diet-induced hyperlipidemia was significantly improved by fermented garlic and orlistat, which includes triglycerides (F = 2878.961; *p* < 0.01), cholesterol (F = 8146.06; *p* < 0.01) high-density lipoprotein cholesterol (HDL) (F = 455.13; *p* < 0.001), low-density lipoprotein cholesterol (LDL) (F = 400.386; *p* < 0.001), and non-HDL cholesterol (F = 473.440; *p* < 0.001). Further post-hoc analysis showed that cholesterol was significantly lower in group I compared to group II, which was diseased. Triglycerides, LDL cholesterol, and non-HDL cholesterol were significantly lower in group I compared to group II. HDL cholesterol was significantly higher in group I compared to group II. Group III, group IV, and group V showed significant improvement in HDL cholesterol after treatment.

### 3.3. Liver Function Parameters

The liver function parameters are shown in Table 3. One-way analysis showed a significant effect of the treatment on bilirubin (F = 8.451; *p* < 0.01), ALT (F = 2293.86; *p* < 0.01), AST (F = 9814.63; *p* < 0.01), alkaline phosphate (F = 7066; *p* < 0.01), protein (F = 2446.7; *p* < 0.01), albumin (F = 7066.038; *p* < 0.01), and the AG ratio (F = 2902.70; *p* < 0.01). Further, the Dunnett test showed that the values of AST, ALT, alkaline phosphate, protein albumin, and the AG ratio of group II were highly significant compared to group I. Treatment groups III and IV showed a significant decrease in values compared to group II, and proved liver function parameters. Group V also showed improvement in liver function parameters compared to the group with high-fat diet-induced obesity.

### 3.4. Insulin

One-way analysis showed significant effect of treatment on the insulin levels (F = 234.52; *p* < 0.01) of the groups. Further Dunnett test analysis proved that group V showed highly significant (*p* < 0.01) improvement in insulin levels compared to group I. Further, group III and group IV showed significant (*p* < 0.00) improvement in insulin levels compared to group I. Group II’s insulin levels were significantly (*p* < 0.00) lower due to obesity in the animals compared to group I as it is mentioned in Figure 2.

### 3.5. Serum IL6 and CPR

The serum concentrations of IL-6 and CPR were assessed to determine the effect of fermented garlic and orlistat on the animal model with high-fat diet-induced obesity. One-way analysis confirmed the statistical effects of different treatments on serum IL-6 (F = 280.15; *p* < 0.01) and CPR (F = 137.88; *p* < 0.01) in the groups. Further analysis showed that CPR and serum IL-6 were significantly (*p* < 0.00) higher in group II compared to group I. Group V showed significant improvement in their inflammatory markers. Those of group III and group IV were also significantly improved compared to Group II as is is mentioned in Figure 3.

### 3.6. Histopathological Slides

Figure 4 shows histopathological changes in the adipose tissues of the experimental groups. In group I, a normal adipocyte number and size were observed. It can also be noted that there was a thin line membrane with clear cytoplasmic lipids. The cell nucleus was also pushed against the cell membrane. In group II, effects such as massive fatty changes, vacuolization, and fatty accumulation in adipose tissues were seen. It was also noted that the cell membrane accumulated fat. There was saturation in the cytoplasmic lipid, indicating by a reddish color. All of these pathological alterations were reduced by fermented garlic an orlistat treatment in group III, group IV, and group V. After six weeks of treatment, there was lower accumulation of fat in the cell membranes, and it was noted that lipid cells were also clearer compared to group II. Group IV and group V showed more improvement in histopathological changes compared to group III.

## 4. Discussion

In addition to increasing the lifelong risk of life-threatening diseases such hypertension, stroke, dyslipidemia, cardiovascular disease, and type 2 diabetes, obesity is a global public health issue [22]. Weight loss can be achieved through a variety of methods, such as lifestyle modifications (diet and exercise), behavioral therapy, medication, and surgery. There is a need for medication as an addition to lifestyle changes in these patients, since short-term achievements in reducing obesity through lifestyle adjustments are frequently insufficient [23]. The first nutritional intervention to control obesity by inducing a high-fat diet in an animal model was invented in 1959 [24]. In the present study, significant increases in weight and organ weight, and elevated levels of lipids, except HDL, were seen after 6 weeks —along with reduced liver function parameters with insulin and inflammatory markers—in animals fed a high-fat diet compared to a control group. Histopathological changes were also noted in the high-fat diet-fed animal model, including significant changes in lipid cell size, a thick line of the cell membrane, and fat accumulation in the adipose tissue. According to studies, body weight gain plays a significant role in the progression of oxidative stress. HFD is associated with increased NADPH oxidase production and the reduced expression or activity of antioxidant enzymes, which results in oxidative stress [25]. In the present study, the histopathological change in adipose tissue is also related to oxidative stress and elevated levels of lipid peroxidation. Typically, obesity is also related to the consumption of a high-fat diet in the long term in animal models with same the clinical problems as obese patients. Both the FDA and the European Medicines Agency (EMA) have authorized orlistat for the treatment of persistent obesity. Although orlistat has been used to treat obesity, reports of gastrointestinal problems are common [26]. Few occurrences of pancreatitis with and without increased amylase, transitory liver failure, severe liver disease, cholestatic hepatitis, tubular necrosis secondary to oxalate nephropathy, myopathy, and hepatocellular necrosis have been documented in the context of orlistat therapy [27]. A case report on thrombocytopenia and macrocytic anemia was recently published [28]. In our study, orlistat significantly improved body weight and lipid profiles, and liver function parameters with insulin levels. It also led to significantly better outcomes in IL-6 and CPR.

Research on black garlic’s anti-inflammatory, immune-boosting, and anti-oxidative properties has focused on these issues [29]. Inflammatory disorders associated with obesity, such as hyperlipidemia, hypertension, arteriosclerosis, type 2 diabetes mellitus, cancer, respiratory issues, and osteoarthritis, are significantly influenced by dietary fat as a factor [30]. In rats with HFD-induced obesity, we looked at the impact of black garlic on growth parameters, serum biochemical parameters, organ and adipose tissue weights, histology, and the antioxidant defense system. According to numerous studies, feeding mice and rats a high-energy diet that contains 30% lard causes the animals to become obese [31]. Fermented garlic powder was given as a supplement with two different dosages for 6 weeks. The dosages were calculated according to the human equivalent dosage. Our data indicated that fermented garlic consumed over weeks suppressed body weight, liver weight, adipose tissue, soleus muscle, and gastrocnemius muscle compared to a high-fat diet. Our data are related to a study in which aged black garlic extract was used as a supplement in a high-fat diet-fed animal model [32]. Obesity is linked to a high incidence of steatosis, including non-alcoholic fatty liver disease (NAFLD) [33]. Reactive oxygen species are primarily produced in fatty liver disease due to the over-accumulation of hepatic lipids and the oxidation of fatty acids [34]. Fermented garlic supplementation resulted in microvascular fat accumulation with small lipid cell sizes in adipose tissue in the HFD groups.

Previous research demonstrated that a black garlic-supplemented HFD led to considerably lower blood triglyceride, AST, and ketone body levels. In addition, an HFD group supplemented with black garlic had considerably higher serum levels of HDL cholesterol than the HFD group [35]. Our data showed a significant effect of fermented garlic on lipid profiles with liver function parameters in rats with high-fat diet-induced obesity. They also showed significant improvement in insulin levels. It has been shown that obesity has harmful impacts on health and negatively affects plasma lipid levels, including high triglycerides and low HDL cholesterol levels [36]. Obesity is linked to decreased antioxidant activity and increased oxidative stress in plasma and organ tissues [37]. Our data also showed a significant effect of fermented garlic on liver function parameters, as it improved AST, ALT, alkaline phosphate, total protein, bilirubin, and the AG ratio in high-fat diet-induced obesity groups treated with fermented garlic and orlistat. This is related to previous research in which rodents specifically exhibited liver injury, as indicated by the activity of serum AST and ALT enzymes [38]. It appears from our data that fermented garlic reduced weight gain, with histopathological changes and improvements in blood biochemical analysis, in combination with the drug orlistat. So, this model could be used to minimize the long-term side effects related to orlistat.

## 5. Conclusions

Rats fed fermented garlic with a high-fat diet had suppressed body weight, liver tissue weight, and soleus and gastrocnemius muscle weight, and improved serum lipid profile levels, liver function parameters with insulin, and IL-6 and CRP. We found that the orlistat and fermented garlic outcomes were close to each other, especially a dosage level of 963.3 mg/kg of fermented garlic. This research also proves the scientific effectiveness of fermented garlic as a therapeutic agent in obesity-caused disfunction and disturbance, and may help reduce the side effects of orlistat. However, further studies are required to investigate the underlying mechanisms related to fermented garlic.

## Figures and Tables

**Figure 1 foods-12-03905-f001:**
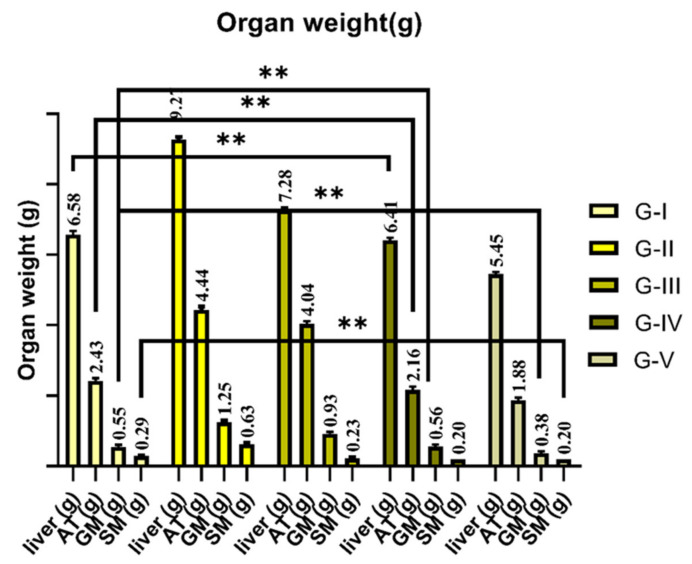
Body weight gain of rats fed a normal and high-fat diet and treated with fermented garlic and orlistat. Data are represented as mean values; *n* = 8 rats/group. ** *p* < 0.01 vs. control. G-I (control), G-II (high-fat diet), G-III (high-fat diet + fermented garlic powder), G-IV (high-fat diet + fermented garlic powder), and G-V (high-fat diet + orlistat). AT—adipose tissue, SM—soleus muscle, and GM—gastrocnemius muscle.

**Figure 2 foods-12-03905-f002:**
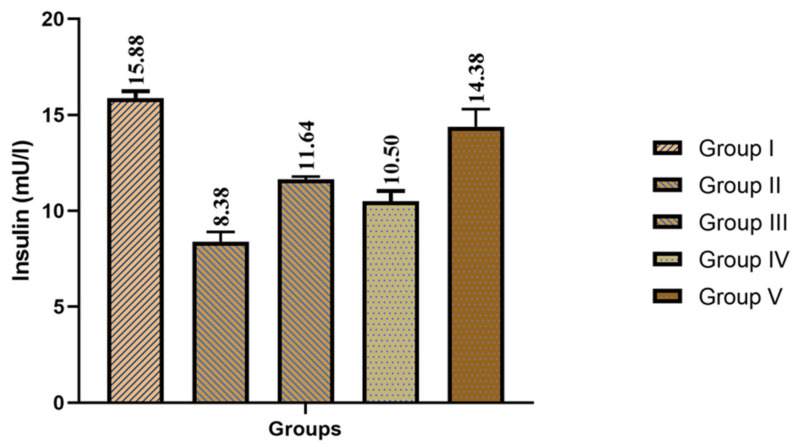
Effect of fermented garlic and orlistat on insulin levels of normal diet- and high-fat diet-fed groups. Data are presented as mean values; *n* = 8 rats/group. Group I (control), Group II (high-fat diet), Group III (high-fat diet + fermented garlic powder), Group IV (high-fat diet + fermented garlic powder), and Group V (high-fat diet + orlistat).

**Figure 3 foods-12-03905-f003:**
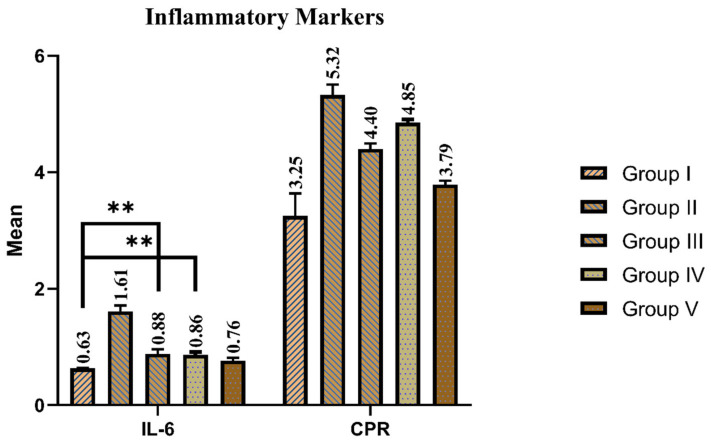
Effect of fermented garlic and orlistat on inflammatory markers of normal diet- and high-fat diet-fed groups. Data are presented as mean values; *n* = 8 rats/group. ** *p* < 0.01 vs. control. Group I (control), Group II (high-fat diet), Group III (high-fat diet + fermented garlic powder), Group IV (high-fat diet + fermented garlic powder), and Group V (high-fat diet + orlistat).

**Figure 4 foods-12-03905-f004:**
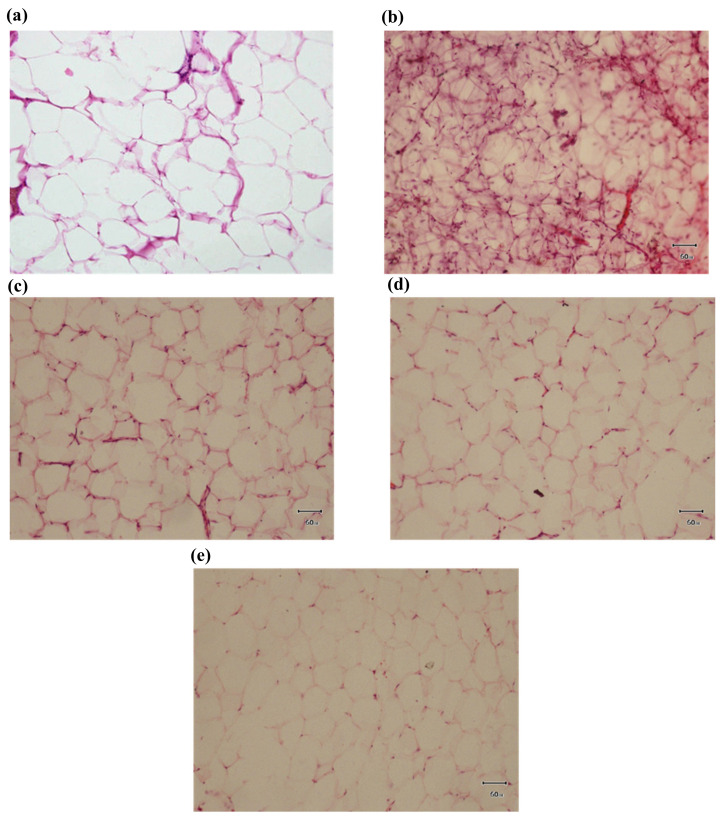
Effect of fermented garlic and orlistat dosage on histomorphological variations in adipose tissues of rats with HFD-induced obesity. (**a**) In group I (control), normal fat cells in adipose tissue were observed. (**b**) In group II (animals with HFD-induced obesity), enormous amounts of fat were stored in cells, and accretion in adipose tissue was detected. (**c**) In group III (HFD + fermented garlic level 1), lower fat deposition with thin cell line membrane were also noted in adipose tissue. (**d**) In group IV (HFD + fermented garlic level 2), lessened fat deposits in tissues were noted. (**e**) In group V (HFD + orlistat), less fat deposition was indicated in adipose tissue.

**Table 1 foods-12-03905-t001:** Effect of fermented garlic and orlistat on body weights of high-fat diet-fed rats.

Groups	1st Week	2nd Week	3rd Week	4th Week	5th Week	6th Week
G-I	250.25 ± 2.12 ^3^	258 ± 0.53 ^3^	264.87 ± 1.89 ^3^	274.12 ± 1.25 ^3^	281.87 ± 0.99 ^3^	291.62 ± 0.74 ^3^
G-II	250.37 ± 0.51 ^4^	262.75 ± 0.16 ^4^	275.75 ± 1.28 ^4^	283.75 ± 1.00 ^4^	296.12 ± 2.64 ^4^	303.75 ± 2.12 ^4^
G-III	249.75 ± 1.98 ^1^	252.87 ± 1.24 ^1^	259.75 ± 1.03 ^1^	266.87 ± 0.99 ^1^	270.75 ± 0.88 ^1^	276.75 ± 1.48 ^1^
G-IV	250.25 ± 2.12 ^2^	255.5 ± 1.19 ^2^	261 ± 0.75 ^2^	269.37 ± 0.51 ^2^	277.5 ± 0.92 ^2^	282.75 ± 0.88 ^2^
G-V	249.62 ± 0.91 ^1^	253.87 ± 0.83 ^1^	258.12 ± 0.83 ^1^	263.62 ± 0.74 ^1^	269 ± 0.75 ^1^	275.375 ± 0.91 ^1^

The values are presented as mean ± SD (*n* = 8). Values indicated with different lower-case numbers in superscript format indicate significant differences at *p* value < 0.01 according to Duncan’s multiple range test; values marked with the same lower-case numbers in superscript format indicate no significant differences. G-I (control), G-II (high-fat diet), G-III (high-fat diet + fermented garlic powder), G-IV (high-fat diet + fermented garlic powder), and G-V (high-fat diet + orlistat).

**Table 2 foods-12-03905-t002:** Effect of fermented garlic and orlistat on lipid profiles of high-fat diet-fed rats.

Groups	Cholesterol (mg/dL)	Triglycerides (mg/dL)	LDL Cholesterol (mg/dL)	HDL Cholesterol (mg/dL)	Non-HDL Cholesterol (mg/dL)
G-I	76 ± 1.06 ^1^	92 ± 2.00 ^1^	69.88 ± 1.45 ^1^	25.13 ± 0.35 ^5^	37.88 ± 1.80 ^1^
G-II	162.88 ± 1.88 ^5^	152.5 ± 2.07 ^4^	91.13 ± 1.24 ^5^	17.25 ± 0.46 ^1^	62 ± 1.92 ^5^
G-III	85.25 ± 1.03 ^4^	97.75 ± 0.88 ^3^	84.38 ± 0.51 ^4^	19.5 ± 0.53 ^2^	56 ± 0.01 ^4^
G-IV	80.75 ± 0.88 ^3^	94.38 ± 0.51 ^2^	81.75 ± 1.28 ^3^	21.5 ± 0.53 ^3^	53 ± 0.01 ^3^
G-V	77.63 ± 0.51 ^2^	91.5 ± 0.53 ^1^	76.5 ± 0.92 ^2^	24 ± 0.01 ^4^	48 ± 0.01 ^2^

The values are presented as mean ± SD (*n* = 8). Values indicated with different lower-case numbers in superscript format indicate significant differences at *p* value < 0.01 according to Duncan’s multiple range test; values marked with the same lower-case numbers in superscript format indicate no significant differences. G-I (control), G-II (high-fat diet), G-III (high-fat diet + fermented garlic powder), G-IV (high-fat diet + fermented garlic powder), and G-V (high-fat diet + orlistat).

**Table 3 foods-12-03905-t003:** Effect of fermented garlic and orlistat on liver function parameters of high-fat diet-fed rats.

Groups	Bilirubin (mg/dL)	ALT (U/L)	AST (U/L)	Alkaline Phosphate (U/L)	Protein (g/dL)	Albumin (g/dL)	AG Ratio
G-I	0.06 ± 0.01 ^1^	35.5 ± 0.53 ^1^	97.75 ± 0.70 ^1^	198.88 ± 0.64 ^1^	5.25 ± 0.5 ^1^	3.03 ± 0.00 ^1^	1.15 ± 0.00 ^2^
G-II	0.32 ± 0.08 ^2^	61.88 ± 0.99 ^5^	171.38 ± 0.74 ^5^	284.5 ± 2.26 ^5^	9.2 ± 0.15 ^5^	5 ± 0.10 ^3^	2.62 ± 0.07 ^3^
G-III	0.32 ± 0.41 ^2^	46.75 ± 0.46 ^4^	141 ± 0.75 ^4^	229.38 ± 0.51 ^4^	7.84 ± 0.05 ^4^	4.0 ± 0.05 ^2^	1.09 ± 0.00 ^1^
G-IV	0.07 ± 0.01 ^1^	42.75 ± 0.46 ^3^	138.62 ± 0.51 ^3^	220.75 ± 0.88 ^3^	7.46 ± 0.05 ^3^	3.99 ± 0.10 ^2^	1.079 ± 0.00 ^1^
G-V	0.59 ± 0.27 ^1^	42 ± 0.01 ^2^	127.13 ± 0.99 ^2^	200 ± 0.53 ^2^	6.41 ± 0.07 ^2^	4.01 ± 0.05 ^2^	1.059 ± 0.01 ^1^

The values are presented as mean ± SD (*n* = 8). Values indicated with different lower-case numbers in superscript format indicate significant differences at *p* value < 0.01 according to Duncan’s multiple range test; values marked with the same lower-case numbers in superscript format indicate no significant differences. G-I (control), G-II (high-fat diet), G-III (high-fat diet + fermented garlic powder), G-IV (high-fat diet + fermented garlic powder), and G-V (high-fat diet + orlistat).

## Data Availability

The research data can be available on request and corresponding author can be contacted for the availability of the research data.

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
