# Peer review of "Comparison of Efficacy of Fermented Garlic and Orlistat (Lipase Inhibitor) in Obesity Management Using an Experimental Rodent Model"

_foods, 2023, doi:10.3390/foods12213905_

Round 1

Reviewer 1 Report

Comments and Suggestions for Authors

The author should add the determination of chemical composition and specific data of black garlic in the article. In addition, the in vitro activity evaluation of black garlic extract is a reasonable supplement to the in vivo experiment. In the introduction part, Taiwan should be amended to Taiwan (China).

Reviewer 2 Report

Comments and Suggestions for Authors

This article is very important. Infromation are originaly but their presentation is not perfect. Designation of groups are by Chapter 2.4 not significantly by Group III, Group IIII and Group IV.  Mistakes between groups continues in chapter 3 and in tables 1, 2, 3, and in figures 1, 2, 3. In the figure 4 are Group III and IV described as HFD + black garlic level 1 (Group III) and HFD + black garlic level 2 (Group IV). This type of description is not in any other chapters. Authors must correcting these inaccuracies by all chapters, tables and figures.

Reviewer 3 Report

Comments and Suggestions for Authors

First Excuse me very much for me Englisch pronounce.

The Thema is good chosen, it is  very interesting to provide some novelty in the field of usfull nutraceuticals, which can be helpfull  in therapeutic way in antiobesity managment , to be some instead od medications, and to evitate the side effects of them. the Manuscript is well designed, bet need to be better structured, apstact, refrence in coparation with the Instructions for Authors, There are some Tipps folowing.

The abstract can be better structured. it is important to be schorter, now it have 243 word counts, but it is necessery to change in schorter form with  200words.  Also must be better structured : Background, Methods, Results, Conclusion. In conclusion must be pointed out the best imapkt  of nutraceutical therapy and underline  novelty. There are some paar corrections, but results and conclusion must be concise and underline the best imapkt and novelty.

there are some changes , but authors need to finisch in 200 words and to pointed betetr resuls and conclusion

Background:Black garlic also known as fermented garlic, is a useful food that may have therapeutic benefits.The aim of this study was to analyze impact of the fermented garlic and orlistat therapy in obese rats. Methods:40 male albino rats (245-250g) were fed either HFD (n=32) or normal diet (n=8) for 6 weeks and therefore randomized (group I) normal diet, (group II) HFD, (group III & IV) HFD with fermented garlic, (group V) orlistat for 6 weeks. Two different dosages of fermented garlic 481.2mg/kg and 963.3mg/kg were administered. Afterward blood was collected, body weight was measured, and tissue was collected for further analysis . Results: The significant (PË‚0.05) low level of insulin with a high level of IL-6 and CPR was noted in HFD induced rats as compared to the control. Further high levels of AST, ALT, AP, total protein, bilirubin, and AG ratio in HFD induced diseased rats as compared to control.The significantly (PË‚0.05) increased body and organ weight with a high level of triglycerides, cholesterol, LDL- cholesterol with decreased HDL-cholesterol in HFD induced rats as compared to the control. Conclusion: must be better structured schort pointed out and novelty, pointed in which group was the lipid and obesity parameter decresed, also in which gropu was the best histopatological changes?

The fermented garlic and orlistat reinstated all those parameters significantly (PË‚0.05) especially body weight and lipid profile and histopathological changes as compared to the drug orlistat. Also it schows antiobesity therapeutic impacts in comparing with an orlistat drug.

In the text the references must be numbered in paraenthesis like this [1].In the text of manuscript the Authors are written (Raheen, Sultam& Yasmeen, 2002) , need to be changed as [1]. On  the end of text reference need to be change not in alfabetical order, insted neeedd to be structured with the number in colerartion with text, while need to be counsalting the instructions for Authors for Journal ~Foods~.

References should be described as follows, depending on the type of work:

  • Journal Articles:
    1. Author 1, A.B.; Author 2, C.D. Title of the article. Abbreviated Journal Name YearVolume, page range.
  • as example 
  • Buckley, L.F.; Carbone, S.; Aldemerdash, A.; Fatani, N.; Fanikos, J. Novel and emerging therapeutics for primary prevention of cardiovascular disease. Am. J. Med. 2019132, 16–24. [Google Scholar] [CrossRef] [PubMed]
  • It is good that the most of reference are up to date within 5 years.
  • in the text need to be specificate concrete in part experimental design is 2 mal repeat group III and  group IIII (high fat diet+fermented garlic powder). My opinion is that is not good understand written the difference between these groups.
  • Discussion is well written, insted of refrence, conclusion also is better written as abstract, while it pointed out importnant facts.
